# Sevelamer Use and Mortality in People with Chronic Kidney Disease Stages 4 and 5 Not on Dialysis

**DOI:** 10.3390/jcm12247631

**Published:** 2023-12-12

**Authors:** Pablo Molina, Mariola D. Molina, Juan J. Carrero, Verónica Escudero, Javier Torralba, Cristina Castro-Alonso, Sandra Beltrán, Belén Vizcaíno, Mercedes González-Moya, Julia Kanter, Asunción Sancho-Calabuig, Jordi Bover, José L. Górriz

**Affiliations:** 1Department of Nephrology, Hospital Universitari Dr. Peset, Fundación para el Fomento de la Investigación Sanitaria y Biomédica de la Comunitat Valenciana, 46017 Valencia, Spain; veesque@gmail.com (V.E.); cristina.c.med@gmail.com (C.C.-A.); sanbelca@gmail.com (S.B.); belvizcaino@gmail.com (B.V.); mgonzalezm@alumni.unav.es (M.G.-M.); julikanter@gmail.com (J.K.); asanchoc2@gmail.com (A.S.-C.); 2Department of Medicine, Universitat de València, 46010 Valencia, Spain; jlgorriz@gmail.com; 3Department of Mathematics, Universidad de Alicante, 03690 Sant Vicent del Raspeig, Spain; mariola.molina@ua.es; 4Department of Medical Epidemiology and Biostatistics, Karolinska Institutet, SE-171 77 Stockholm, Sweden; juan.jesus.carrero@ki.se; 5Department of Nephrology, Hospital General Universitario, 03010 Alicante, Spain; torralba_fra@gva.es; 6Nephrology Department, University Hospital Germans Trias i Pujol, 08916 Badalona, Spain; jbover.ics@gencat.cat; 7Department of Nephrology, Hospital Clínico Universitario, Fundación para la Investigación del Hospital Clínico de la Comunidad Valenciana, 46010 Valencia, Spain

**Keywords:** calcium-free phosphate binders, chronic kidney disease, hyperphosphatemia, phosphate, sevelamer, survival analysis

## Abstract

*Rationale and objective:* Data suggest that non-calcium-based binders, and specifically sevelamer, may lead to lower rates of death when compared with calcium-based binders in end-stage renal disease (ESRD) patients. However, the association between sevelamer use and mortality for those with non-dialysis-dependent chronic kidney disease (NDD-CKD) patients has been uncertain. *Study design:* Our research is presented in a prospective cohort study. *Setting and participants:* A total of 966 participants with NDD-CKD stages 4–5 were enrolled in the PECERA study from 12 centers in Spain. *Exposure:* The participants were treated with sevelamer. *Outcome:* This study yielded all-cause and cardiovascular mortality outcomes. *Analytical approach:* We conducted an association analysis between mortality and sevelamer use with time-dependent Cox proportional hazards models. *Results:* After a median follow-up of 29 months (IQR: 13–36 months), death occurred in 181 participants (19%), with cardiovascular (*n* = 95, 53%) being the leading cause of death. In a multivariable model, the adjusted hazard ratios (HRs) for patients under sevelamer treatment were 0.44 (95% CI, 0.22 to 0.88) and 0.37 (95% CI, 0.18 to 0.75) for all-cause and cardiovascular mortality, respectively, compared with those of untreated patients. *Limitations:* Some limitations include potential confusion via indication bias; causal statements about these associations cannot be made due to the observational nature of this study. *Conclusions*: In this prospective NDD-CKD cohort study, the administration of sevelamer was independently associated with lower all-cause and cardiovascular mortality, suggesting that non-calcium-based phosphate binders might be the first-line therapy for phosphate lowering in this population. Further interventional studies clarifying the risks and benefits of phosphate binders in NDD-CKD are warranted.

## 1. Introduction

As chronic kidney disease (CKD) progresses, phosphaturic mechanisms such as fibroblast growth factor 23 (FGF23) and parathyroid hormone (PTH) secretion gradually become unable to overcome the continuous supply of phosphate from dietary intake, leading to a positive phosphate balance and hyperphosphatemia [1,2]. Elevations in serum phosphate concentration occur in patients with advanced CKD, especially after progression to stages 4 and 5, representing a risk factor for mortality in people with non-dialysis-dependent CKD (NDD-CKD) [3,4,5,6,7]. Therefore, it is possible, although not yet fully proven, that interventions aimed at reducing serum phosphate levels may improve survival in this population [8,9,10].

Dietary phosphate restriction is the first step in the management of hyperphosphatemia in CKD [11,12,13]. However, a low phosphate diet is difficult to follow and usually insufficient to achieve adequate control of serum phosphate [10]. As a result, phosphate binders represent the most widely applied treatment for hyperphosphatemia in daily clinical practice, with the calcium content of the binders being their defining characteristic [9]. Calcium-based binders include calcium acetate and carbonate, whereas non-calcium-based binders include aluminum hydroxide, magnesium carbonate, sevelamer, lanthanum carbonate, sucroferric oxyhydroxide and ferric citrate [13]. Calcium-based binders, unlike non-calcium-based binders, may contribute to calcium accumulation due to them being associated with hypercalcemia, adynamic bone disease and vascular calcification, all of which could result in increased mortality [14,15,16,17,18].

Although non-calcium-based binders, and specifically sevelamer, may lead to lower death when compared with calcium-based binders in end-stage renal disease (ESRD) patients on dialysis [8,16,19], clinical trials have not definitively demonstrated the superiority of any class of phosphate binders over another with respect to clinical outcomes in NDD-CKD patients [8]. Moreover, the lack of placebo-controlled trials examining the impact of phosphate binders on hard endpoints such as survival or quality of life has hampered our ability to apply phosphate binders as a means of achieving clinically relevant benefits beyond serum phosphate reduction [9,10]. As a result, current clinical practice guidelines, while recognizing low-level evidence, do not expressly address significant considerations regarding the use of phosphate binders and suggest “restricting the dose of calcium-based phosphate binders” in CKD stage 3 through ESRD [11,20,21]. In this absence of randomized evidence, observational effectiveness studies may be useful [9].

Awareness of the lack of evidence behind guideline recommendations, coupled with our concerns about calcium exposure from phosphate binders and hyperphosphatemia risks, encouraged us to investigate the association of non-calcium-based phosphate binder administration with mortality. The goal of our study was to determine the patterns of use of non-calcium-based phosphate binders and their associated death in a prospective cohort of stage 4–5 NDD-CKD patients.

## 2. Materials and Methods

### 2.1. Study Design and Patient Selection

PECERA (Collaborative Study Project In patients with Advanced Renal Failure) was a prospective, observational, multi-center study of patients with CKD stage 4 or 5 not receiving dialysis attending 12 centers in the Valencian Community of Spain to evaluate the ability of calcium, phosphate and PTH levels to predict death. Details about the design and main results of the PECERA study have previously been published [7]. Briefly, stage 4–5 NDD-CKD patients were included and prospectively followed for 3 years. Patient data were collected at baseline and 6, 12, 24 and 36 months after the enrollment. Treatment variables (which included, among other ongoing medications, calcium-based and non-calcium-based phosphate binders), comorbidity history as well as a clinical assessment and laboratory parameters were recorded during the 6-month observations. In each follow-up examination, patient status and hospitalizations in the preceding six months were also thoroughly documented. Follow-up continued until death, commencement of renal replacement therapy, loss to follow-up or 36 months from study entry.

This study was reviewed and approved by the Dr Peset Hospital Research Ethics Committee (approval code 47/06, approval date 30 June 2006) and was conducted in agreement with the Declaration of Helsinki. All participants provided informed written consent.

### 2.2. Study Exposure

In this post hoc analysis of the PECERA study, the study exposure was the treatment with non-calcium-based phosphate binders as a time-dependent variable, with repeated measures at baseline, 6, 12, 24 and 36 months. Sevelamer was the only non-calcium-based binder used, whereas the calcium-based binders include calcium acetate and carbonate.

### 2.3. Study Outcomes

All-cause mortality and cardiovascular mortality were the primary and secondary outcomes, respectively. Cardiovascular mortality included any death secondary to stroke, heart failure, myocardial ischemia or peripheral vascular disease. 

### 2.4. Study Covariates

Covariates tested included sex, age, comorbidities, cause of CKD, body measurements and laboratory parameters, as previously described [7]. The history of cardiovascular disease was defined according to the presence of coronary artery disease, chronic heart failure, cerebrovascular disease or peripheral vascular disease prior to enrollment. Additional comorbidities registered in our study forms included hypertension, diabetes mellitus and smoking status. Hypertension was defined as SBP  ≥  140 mm Hg and/or DBP  ≥  90 mm Hg or the current use of antihypertensive agents. Diabetes was defined as HbA1c  ≥  6.5% and/or the current use of antihyperglycemic agents. Estimated glomerular filtration rate (eGFR) was determined according to the CKD-EPI formula [22]. In addition to phosphate binders, treatment with antihypertensive drugs, iron, erythropoiesis-stimulating agents (ESAs), and both native and active vitamin D in each study visit were also included as covariates. 

### 2.5. Statistics

Summary statistics were reported as frequencies or percentages and as mean ± SD for categorical and quantitative variables, respectively. Skewed quantitative variables were expressed as the median (interquartile range). Differences in the characteristics of patients administered sevelamer versus patients who did not were assessed with the t-test or with the Mann–Whitney U test for continuous variables, where appropriate, and the chi-squared test for categorical variables.

Survival curves and the log-rank test were used to estimate the effects of sevelamer exposure on all-cause mortality. The independent association between sevelamer treatment on all-cause and cardiovascular mortality was assessed with time-dependent Cox proportional hazard regression analyses. The exposure variables (treatment with sevelamer at each visit) were introduced in the Cox models as categorical variables, whereas the laboratory parameters were introduced as continuous variables and fitted using non-linear p-splines [23].

For the multivariate model building, we first included age, sex, comorbidities, cause of CKD, and anthropometric measurements (Model 1). Model 2 included concomitant medications plus those variables in Model 1. Lastly, we generated a fully adjusted model (Model 3), including all previous variables plus laboratory parameters. Because the analysis utilized repeated measurements of each individual, all variables included were modeled as time-varying throughout all six-month patient visits, including the patient identifier as a cluster variable to account for correlated observations within each patient. All Cox models were stratified by center. 

The literature indicates that annual mortality in patients with NDD-CKD stages 4 to 5 is around 8% [24]. Previous studies have shown a 45% prevalence of hyperphosphatemia [4]. Compared with the group with normal phosphate levels, the group with hyperphosphatemia shows an 80% increase in mortality [3]. With 966 patients included, assuming losses of 20% and considering three years of follow-up and an error of alfa = 0.05, the power estimation of this study is 0.8. All statistical analyses were performed using R statistical software version 4.3.1 with the “survival” and “R commander” packages (R Foundation for Statistical Computing, Vienna, Austria) [23].

## 3. Results

### 3.1. Patient Characteristics Using Sevelamer Treatment

A total of 966 patients (women: 39%; mean age: 69.6 ± 13.7 years) were included in this pre-specified analysis. At baseline, 707 (73%) and 259 (27%) patients had CKD stages 4 and 5, respectively (Table 1). Overall, 515 (53%) patients received some form of binder, with most of them using calcium-based binders (*n* = 360, 37%) and a minority (*n* = 111, 11%) using exclusively sevelamer (*n* = 111, 11%) or a combination of the two (*n* = 44, 5%).

At baseline, the proportion of patients with stage 5 CKD was higher in the sevelamer treatment group, which showed lower levels of BMI, eGFR, albumin and C-reactive protein (CRP). Calcium, phosphate and PTH levels were increased in the sevelamer group, in which treatment with vitamin D compounds and diuretics was prescribed more frequently. Conversely, the proportion of patients treated with renin–angiotensin–aldosterone system inhibitors and ESAs was higher among patients without sevelamer treatment (Table 1).

### 3.2. Mortality

A total of 181 (19%) patients died after a median follow-up of 29 months (interquartile range: 13–36 months). Cardiovascular disease (*n* = 95, 52%), infections (*n* = 25, 14%) and tumors (*n* = 20, 11%) were the most common causes of death. Renal death and others accounted for 7% (*n* = 13) and 6% (*n* = 10) of deaths, respectively. In 18 cases (10%), the cause of death was not identified. Survival curves between patients administered sevelamer and patients who were not (Figure 1) suggested a survival benefit for patients on sevelamer treatment (log-rank test; *p* = 0.008). 

In the crude model (Model 0, Table 2), about a half-lower risk of all-cause mortality was found among patients receiving sevelamer compared to patients who did not (unadjusted HR, 0.50 (95% CI, 0.29 to 0.87); *p* = 0.014). Adjustments for demographic, anthropometric characteristics, comorbidities, and medications strengthened this difference (Models 1 and 2, Table 2). The full adjustment model included biochemical characteristics and showed that sevelamer use was associated with significantly lower all-cause mortality (fully adjusted HR, 0.37 (95% CI, 0.18 to 0.75); *p* = 0.006). 

For cardiovascular mortality, minimally and fully adjusted analyses (Models 1, 2 and 3, Table 2) showed that sevelamer treatment was also significantly associated with lower mortality risk (fully adjusted HR, 0.28 (95% CI, 0.12 to 0.67); *p* = 0.005).

## 4. Discussion

In the PECERA study, treatment with sevelamer was associated with lower all-cause and cardiovascular mortality in Spanish stage 4–5 NDD-CKD patients. The better survival rate was observed despite higher percentages of stage 5 CKD, higher serum levels of Ca_alb_, phosphate and PTH, and lower eGFR and albumin levels, factors that have all been associated with higher mortality risk in NDD-CKD [1,3,4,5,6,7,25,26,27,28]. Conversely, other factors related to lower mortality risk, such as lower levels of CRP and more use of vitamin D compounds, were also observed in the sevelamer-treated group [26,29,30]. Due to the awareness of the known effects of all these confounding factors and their potential interactions, appropriate multivariate adjustments were performed [31].

The benefits of survival associated with the use of non-calcium-based binders in CKD patients have been addressed in previous studies and several meta-analyses. Whereas several studies and some systematic reviews of currently available data suggest a potential benefit of calcium-free phosphate binders, and specifically sevelamer, over calcium-based binders for overall survival in dialysis patients [8,16,19,32,33,34,35], the results of these studies remain controversial in the NDD-CKD population [8,9,10,16,19,36]. In a systematic meta-analysis of 11 open-label, randomized trials that included 4622 patients and 936 deaths, there was a 22% decrease in all-cause mortality in stage 3–5D CKD patients randomly assigned to receiving non-calcium-based phosphate binders (sevelamer, 10 studies including 3268 patients, or lanthanum, 1 study including 1354 patients) compared with patients assigned to calcium-based binders (RR 0.78, 95% CI 0.61–0.98) [19]. A subgroup analysis involving only NDD-CKD patients (two studies, 302 patients) [36,37] attenuated the difference in mortality between groups (RR 0.54, 95% CI 0.28–1.03). A second meta-analysis of 25 randomized clinical trials involving 13,744 adults with stage 3–5D CKD also reported a decrease in all-cause mortality with sevelamer compared with calcium-based binders (13 studies, *n* = 3799, RR 0.54, 95% CI 0.32–0.93) [16]. No subgroup analysis of NDD-CKD patients (three studies, 450 patients) [36,37,38] was performed. More recently, a Cochrane review involving 104 randomized clinical trials with 13,744 patients with CKD concluded that sevelamer may lead to lower all-cause death (RR 0.53, CI 0.30 to 0.91) in CKD G5D when compared with calcium-based binders, whereas non-calcium-based binders (including sevelamer, lanthanum and iron-based phosphate binders) had uncertain or inestimable effects on survival in NDD-CKD stages 2 to 5 [8]. Conversely, in the only randomized clinical trial aimed to evaluate all-cause mortality as the primary endpoint in NDD-CKD patients, Di Iorio et al. [36] observed a lower mortality rate among patients receiving sevelamer vs. calcium carbonate. Differences in study populations, sample size and follow-up are likely reasons for the discrepancy between NDD and dialysis studies [39].

Of note, in all the three already-mentioned meta-analyses, there was no difference between groups in serum phosphate, suggesting that the reduction in mortality associated with non-calcium-based phosphate binders seemed to be independent of the degree of serum phosphate reduction [8,16,19]. The lower risks for calcium accumulation, hypercalcemia, vascular calcification and adynamic bone disease related to non-calcium-based binders are potential mechanisms for explaining their survival benefit [14,15,16,17,18].

Additionally, sevelamer may produce further benefits through many pleiotropic actions on lipid profiles, inflammatory markers, uremic toxins and blood glucose levels in CKD patients [40,41]. Based on its ability to bind bile salts, sevelamer has been demonstrated to reduce low-density lipoprotein cholesterol and plasma glucose levels [42,43,44]. These effects have recently been summarized in a meta-analysis including 44 studies for qualitative analysis and 28 reports for quantitative analysis, in which sevelamer showed a significant reduction in cholesterol levels compared to calcium-based phosphate binders, with a decrease in glycated hemoglobin levels in sevelamer-treated patients [40]. Anti-inflammatory effects for sevelamer-based products have also been well demonstrated. In a randomized, controlled, open-label crossover trial involving 53 patients with NDD-CKD, Ruggiero et al. showed that sevelamer reduced CRP, glycated hemoglobin, and total- and low-density lipoprotein cholesterol levels and increased high-density lipoprotein cholesterol levels without affecting GFR; proteinuria; blood pressure; or levels of fibroblast growth factor 23, klotho, intact parathyroid hormone or serum vitamin D [45]. Decreases in biomarkers of oxidative stress as well as in uremic toxins such as p-cresyl sulfate are additional benefits observed after sevelamer treatment [41,46]. These data are in line with the results of our study, in which we observed a survival benefit in the group of patients who received treatment with sevelamer despite having significantly higher phosphorus levels than those patients who did not. Although it could be argued that higher levels of phosphate could be explained by a higher protein intake, which could justify a better nutritional status and, therefore, a survival benefit, unfortunately, protein and caloric intake were not measured in the PECERA study. We speculate, however, that the higher phosphate levels observed in the sevelamer group were not associated with better nutritional status, given that BMI and albumin levels were slightly but significantly lower in the sevelamer group.

Because of a lack of placebo-controlled studies, questions remain regarding phosphate binder benefits for patients with NDD-CKD. Although sevelamer has the longest experience and the greatest number of clinical data among the non-calcium-based binders, it is noteworthy that all these results have been conducted in comparative trials in which calcium-based binders rather than placebos were used as the control [8,16,19]. Moreover, to the best of our knowledge, no previous studies have examined the effect of phosphate binder use on cardiovascular mortality. In this study, we sought to fill this evidence gap and to compare the survival of patients on sevelamer treatment with those who were not, showing a benefit for all-cause and cardiovascular death after adjusting for calcium-based binder use. These results suggest that the potential mechanisms via which sevelamer may reduce the risk of death include not only the absence of calcium overload [47,48] but also other mechanisms, such as the anti-atherogenic effect mentioned above plus other pleiotropic effects beyond phosphate binding [43,44,49,50]. It is noteworthy that, in our cohort of patients, the use of calcium-based phosphate binders did not predict death (Appendix A).

The strong points of this study include the relatively high number of patients included with low loss of follow-up, the prospective data collection and the 3-year follow-up. In contrast, there are several limitations to be commented on. As an observational study, it is still insufficient to determine whether the association between sevelamer treatment and lower death risk is causal and reversible, which should be tested in future placebo-controlled clinical trials. Because the allocation of treatment (as in any observational study) was not randomized and the indication for sevelamer (i.e., hyperphosphatemia) was related to the risk of death, the resulting imbalance in the underlying risk profile between sevelamer and comparison groups could generate biased results (confounding by indication) [51,52]. We do, however, speculate that this risk bias may be mitigated by the fact that the sevelamer group presented a survival benefit in spite of having a higher risk profile and that 37% of patients in the non-sevelamer group also received phosphate binders. Lastly, it would have been desirable to have data on vascular calcification, given the highest prognostic value of this parameter in the CKD population and its potential relationship with the calcium content of the different phosphate binders [17].

Prospective large and long-term registry approaches are certainly inferior and suboptimal data resources when compared with randomized clinical trials, especially regarding potential cause-and-effect relationships, but they will still provide some valuable information in the NDD stages of CKD [53]. Moreover, observational databases can be useful complements to randomized controlled trials as a means to observe whether the efficacy of controlled conditions in specialized centers translates into effective treatment in routine practice [54].

## 5. Conclusions

In summary, in PECERA, after 3 years of follow-up, a statistically significant association between sevelamer prescription and survival was found in stage 4–5 NDD-CKD patients independent of phosphate levels and Ca-based binder prescription, as previously reported in patients on dialysis. Similar results were observed in the case of cardiovascular mortality. Whether these benefits can be extrapolated to other non-calcium-based binders remains unanswered. Clinical trials are warranted to clarify the risks and benefits of phosphate binders in people with NDD-CKD, with the aim of permitting the selection of the best available therapy for the individual patient.

## Figures and Tables

**Figure 1 jcm-12-07631-f001:**
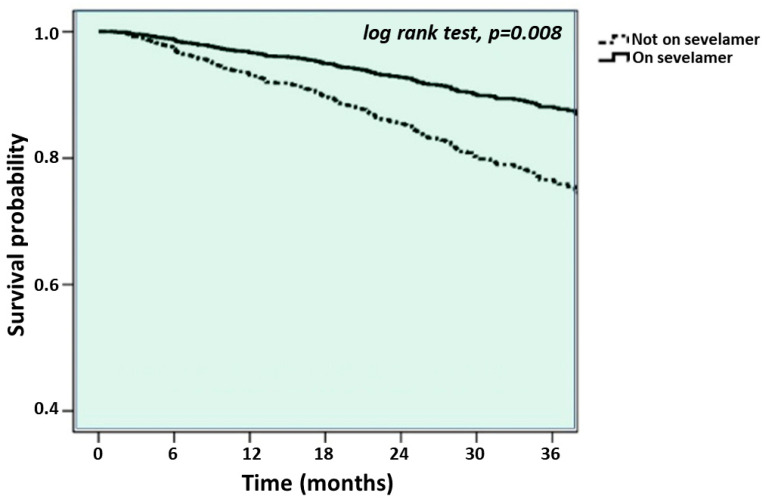
Survival curves between patients administered sevelamer and patients who were not.

**Table 1 jcm-12-07631-t001:** Baseline patient characteristics stratified using sevelamer treatment status (*n* = 966).

Characteristics	N_base_	All (n = 966)	Sevelamer Treatment (n = 155)	No Sevelamer Treatment (n = 811)	*p*
Age (years)	966	69.6 ± 13.7	68.3 ± 14.4	69.9 ± 13.5	0.20
Sex (%)	966				0.11
Men		586 (61%)	85 (55%)	501 (62%)	
Women		380 (39%)	70 (45%)	310 (38%)	
CKD stage (%)	966				<0.001
4		707 (73%)	86 (56%)	621 (77%)	
5		259 (27%)	69 (44%)	190 (23%)	
Cause of CKD (%)	966				0.45
Nephrosclerosis		401 (42%)	66 (43%)	335 (41%)	
Diabetic nephropathy		124 (13%)	18 (12%)	106 (13%)	
Interstitial		103 (11%)	19 (12%)	84 (10%)	
Glomerular		61 (6%)	6 (4%)	55 (7%)	
Polycystic		44 (5%)	11 (7%)	33 (4%)	
Other causes		120 (12%)	16 (10%)	104 (13%)	
Not specified		113 (12%)	19 (12%)	94 (12%)	
Cardiovascular disease history	966	191 (20%)	36 (23%)	155 (19%)	0.27
Diabetes mellitus (%)	966	346 (36%)	54 (33%)	292 (36%)	0.86
Smoking (%)	966				0.03
Never		531 (55%)	95 (61%)	436 (54%)	
Ex-smoker		330 (34%)	29 (25%)	291 (36%)	
Active		105 (11%)	21 (14%)	84 (10%)	
BMI (Kg/m^2^)	965	28.3 ± 5.3	27.2 ± 5.1	28.5 ± 5.3	0.005
BP (mm Hg)	966				
Systolic		132.7 ± 17.3	131.5 ± 14.9	132.9 ± 17.7	0.28
Diastolic		72.8 ± 9.4	71.3 ± 9.9	72.6 ± 9.0	0.49
Waist circumference (cm)	963	100.9 ± 15.3	101.3 ± 15.5	100.8 ± 14.8	0.67
eGFR (mL/min/1.73 m^2^)	966	18.6 ± 5.0	16.6 ± 4.8	19.0 ± 4.9	<0.001
24 h proteinuria (g/24 h)	933	0.66 (0.23–1.65)	0.75 (0.28–1.67)	0.64 (0.21–1.60)	0.29
Hemoglobin (g/L)	964	12.3 ± 1.4	12.3 ± 1.3	12.3 ± 1.5	0.49
Albumin(g/dL)	942	4.1 ± 0.4	4.0 ± 0.5	4.1 ± 0.4	0.004
Intact parathyroid hormone (pg/mL) ^a^	936	133 (85–208)	164 (105–252)	130 (80–197)	<0.001
Phosphate (mg/dL)	965	4.0 ± 0.7	4.2 ± 0.8	4.0 ± 0.7	0.007
Ca_alb_ (mg/dL)	966	9.3 ± 0.5	9.6 ± 0.6	9.3 ± 0.5	<0.001
Potassium (mEq/L)	966	4.9 ± 0.6	4.9 ± 0.6	5.0 ± 0.6	0.10
C reactive protein (mg/dL) ^a^	685	2.0 (0.5–5.1)	0.4 (0.3–1.5)	2.4 (1.0–6.4)	<0.001
Bicarbonate (mEq/L)	807	24.1 ± 3.5	24.4 ± 3.7	24.0 ± 3.4	0.19
CKD–MBD therapy (%):	966				
Native vitamin D		65 (7%)	24 (16%)	41 (5%)	<0.001
Active vitamin D		311 (32%)	100 (65%)	211 (26%)	<0.001
Calcium-based phosphate binder		336 (35%)	34 (22%)	302 (37%)	<0.001
Calcium-free phosphate binder		101 (11%)	155 (100%)	-	NA
Antihypertensive drugs (%)	966				
RAAS inhibition (%)		709 (73%)	98 (63%)	611 (75%)	0.005
Diuretic (%)		647 (67%)	118 (76%)	529 (65%)	0.009
Beta blockers (%)		269 (28%)	52 (34%)	217 (27%)	0.10
Other treatments (%)	966				
ESA		427 (44%)	274 (39%)	341 (42%)	0.003
Iron		516 (53%)	352 (50%)	440 (54%)	0.25

Abbreviations: BMI, body mass index; BP, blood pressure; Ca_alb_, calcium corrected for albumin; CKD–MBD, chronic kidney disease–mineral and bone disorder; eGFR, estimated glomerular filtration rate; ESA, erythropoietin-stimulating agent; N_base_, information available at baseline; RAAS, renin–angiotensin–aldosterone system. Quantitative results are presented as mean ± SD, except skewed values ^a^ which are presented as median (interquartile range).

**Table 2 jcm-12-07631-t002:** Hazard ratios (HRs) and 95% confidence intervals (CIs) for the risk of all-cause and cardiovascular mortality, comparing sevelamer-treated vs. untreated patients, unadjusted and after multivariable adjustments for various confounders.

		All-Cause Mortality		Cardiovascular Mortality	
Model no.	Covariates	HR (95% CI)	*p*	HR (95% CI)	*p*
0	Unadjusted	0.50 (0.29–0.87)	0.014	0.51 (0.25–1.04)	0.06
1	Demographic, anthropometric characteristics and comorbidities ^a^	0.51 (0.27–0.94)	0.03	0.43 (0.19–0.96)	0.04
2	Model 1 plus medications ^b^	0.45 (0.24–0.85)	0.014	0.40 (0.17–0.94)	0.04
3 (full)	Model 2 plus laboratory parameters ^c^	0.37 (0.18–0.75)	0.006	0.28 (0.12–0.67)	0.005

Note: The same covariates were included in all of the fully adjusted analyses. ^a^ Model 1 included sex, age, body mass index, waist circumference, diabetes, cardiovascular comorbidity, and systolic and diastolic pressure. ^b^ Model 2 included all previous variables plus treatment with antihypertensive drugs, iron, erythropoiesis-stimulating agents, and both native and active vitamin D. ^c^ Model 3 (full model) was adjusted for those variables included in Model 1 and 2 plus serum levels of phosphate, iPTH, calcium, albumin, eGFR, hemoglobin, potassium, C-reactive protein and potassium, and 24 h proteinuria.

## Data Availability

Data are contained within the article.

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
