# Peer review of "Sevelamer Use and Mortality in People with Chronic Kidney Disease Stages 4 and 5 Not on Dialysis"

_jcm, 2023, doi:10.3390/jcm12247631_

Round 1
Reviewer 1 Report
Comments and Suggestions for Authors
I enjoyed reading this study. It is interesting and relevant.
I have a few comments.
- How was sevelamer (or other calcium binders) exposure evaluated? Did the patients have to be on sevelamer treatment for the entire follow-up period? A minimum period of time? Being only prescribed the phosphate binder once during the follow-up period? Being on a phosphate binder at inclusion no matter the duration of treatment and then being considered intention to treat?
- I think it should be useful to include the treatment time in the regression models as well as the duration of kidney disease.
- - Were vascular calcifications assessed during the study?
- Please defined the comorbidities evaluated and the criteria to diagnose them.
Author Response
ANSWER TO REVIEWER 1 COMMENTS
- Comment 1. I enjoyed reading this study. It is interesting and relevant.
Thank you.
- Comment 2. How was sevelamer (or other calcium binders) exposure evaluated? Did the patients have to be on sevelamer treatment for the entire follow-up period? A minimum period of time? Being only prescribed the phosphate binder once during the follow-up period? Being on a phosphate binder at inclusion no matter the duration of treatment and then being considered intention to treat?
Thank you for your comment. The exposure to sevelamer (as well as to calcium-based phosphate binders) was evaluated using repeated measurements on individuals over time, at 6-monthly intervals. Thus, each patient could be exposed to the diverse treatments in a variable manner in the different 6-month periods throughout the observation time. Because the analysis utilized repeated measurements on each individual, appropriated regression modelling strategies were used (e.g., random effects models for continuous outcomes, generalized estimating equations models for categorical outcomes).
To clarify this, two new paragraphs have been added to Material and Methods section, as follows:
“Treatment variables (which included, among other ongoing medications, calcium-based and non-calcium-based phosphate binders), comorbidity history as well as a clinical assessment and laboratory parameters were recorded during the 6-monthly observations. In each follow-up examination, patient status and hospitalizations in the preceding six months were also thoroughly documented. Follow-up continued until death, commencement of renal replacement therapy, loss to follow-up or 36 months from study entry.”
“Because the analysis utilized repeated measurements on each individual, all variables included were modelled as time-varying throughout all six-monthly patient visits, including the patient identifier as a cluster variable to account for correlated observations within each patient.”
- Comment 3. I think it should be useful to include the treatment time in the regression models as well as the duration of kidney disease.
As previously explained the treatment variables were included and modelled as time-varying throughout all six-monthly patient visits. Conversely, we did not record the duration of CKD, although we think that the absence of this data may be corrected (at least partially) by introducing estimated glomerular filtration rate and proteinuria into the different models, also as time-dependent variables.
- Comment 4. Were vascular calcifications assessed during the study?
Thank you for pointing this out. Unfortunately, data about vascular calcifications were lacking. We have introduced this as a new limitation of the study as follows:
“Lastly, it would have been desirable to have data on vascular calcification given the highest prognostic value of this parameter in CKD population and its potential relationship with the calcium content of the different phosphate binders [17].”
Ref. 17. Górriz JL, Molina P, Cerverón MJ, Vila R, Bover J, Nieto J, Barril G, Martínez-Castelao A, Fernández E, Escudero V, Piñera C, Adragao T, Navarro-Gonzalez JF, Molinero LM, Castro-Alonso C, Pallardó LM, Jamal SA. Vascular calcification in patients with nondialysis CKD over 3 years. Clin J Am Soc Nephrol. 2015 Apr 7;10(4):654-66. doi: 10.2215/CJN.07450714. Epub 2015 Mar 13. PMID: 25770175; PMCID: PMC4386255.
- Comment 5. Please defined the comorbidities evaluated and the criteria to diagnose them.
The history of cardiovascular disease was defined according to the presence of coronary artery disease, chronic heart failure, cerebrovascular disease or peripheral vascular disease prior to enrollment. Additional comorbidities registered in our study forms included hypertension, diabetes mellitus and smoking status. Hypertension was defined as SBP ≥ 140 mm Hg and/or DBP ≥ 90 mm Hg or the current use of antihypertensive agents. Diabetes was defined as HbA1c ≥ 6.5% and/or the current use of antihyperglycemic agents. All this information has been added to the Material and Methods section.
Reviewer 2 Report
Comments and Suggestions for Authors
In this retrospective analysis of the PECERA (Collaborative Study Project in Patients with Advanced Renal Failure) study, encompassing individuals with CKD Stage 4 or 5 not requiring dialysis across 12 centers in the Valencian Community of Spain, the investigation focused on comparing all-cause and cardiovascular mortality concerning the use of non-calcium-based versus calcium-based phosphate binders. Notably, the study included 966 patients, among whom 53% (515 patients) received some form of binder—comprising calcium-based binders (n=360, 37%), exclusively sevelamer (n=111, 11%), or a combination of the two (n=44, 5%).
The primary and secondary outcomes examined were all-cause mortality and cardiovascular mortality, respectively. Over a median follow-up duration of 29 months, the study recorded 181 deaths (19%), with cardiovascular events accounting for the majority (n=95, 53%) of these deaths. Upon adjusting for various factors such as age, weight, blood pressure, diabetes, comorbidity, vitamin D treatment, renal function, and levels of albumin, calcium, phosphorous, and PTH through multivariate analysis, treatment with sevelamer emerged as independently associated with significantly lower rates of both all-cause (adjusted HR, 0.44 (95% CI, 0.22 to 0.88); p=0.02) and cardiovascular mortality (adjusted HR, 0.37 (95% CI, 0.18 to 0.75); p=0.006). The findings of this study propose that in advanced non-dialysis-dependent CKD patients, the use of non-calcium-based phosphate binders, particularly sevelamer, may warrant consideration as the primary choice for phosphate-lowering therapy. This study stands out as one of the few scientific papers dedicated to investigating the impact of Sevelamer on mortality in a significant cohort of CKD patients not reliant on dialysis. The paper exhibits a well-structured and logically organized presentation, including clear delineation of methods, results, and tables, thus holding potential clinical significance.
Comments
1. It might be worth exploring whether a more distinct assessment of mortality could be achieved by comparing a group of patients solely treated with Sevelamer against those receiving calcium-based phosphate binders. Additionally, elucidating the status of the remaining 451 patients—who may have (or have not) received some form of phosphate binder—and their mortality rates could offer further context.
2. Numerous experimental, observational studies, and clinical trials have highlighted Sevelamer's multifaceted effects beyond controlling hyperphosphatemia. These effects encompass actions on inflammation, oxidative stress, lipid profile, atherogenesis, vascular calcification, endothelial dysfunction, and reduction of uremic toxins. Acknowledging these pleiotropic effects in the discussion, along with its ability to bind bile salts—a factor potentially contributing to benefits such as reduced low-density lipoprotein cholesterol and plasma glucose levels (mentioned in lines 247 to 248)—could enrich the overall understanding of Sevelamer's comprehensive impact on cardiovascular morbidity and mortality in chronic kidney disease patients.
Author Response
ANSWER TO REVIEWER 2 COMMENTS
Comment 1. It might be worth exploring whether a more distinct assessment of mortality could be achieved by comparing a group of patients solely treated with Sevelamer against those receiving calcium-based phosphate binders. Additionally, elucidating the status of the remaining 451 patients—who may have (or have not) received some form of phosphate binder—and their mortality rates could offer further context.
The authors appreciate this comment. Before writing the definitive version of the manuscript, we performed several survival analyses comparing the mortality rates between sevelamer vs. those without any form of phosphate binder, calcium-based binders vs. those without any form of phosphate binder, as well as patients receiving sevelamer vs. calcium-based phosphate binders.
As we explained in the Discussion section, although sevelamer has the longest experience and the greatest amount of clinical data among the non-calcium-based binders, it is noteworthy that all these results have been conducted in comparative trials in which calcium‐based binders rather than placebo were used as the control. Because of a lack of these placebo-controlled studies, in this post-hoc analysis of the PECERA cohort we sought to fill this evidence gap, comparing the survival of patients on sevelamer treatment with those who were not, rather to compare it against patients treated with calcium-based binders. The survival benefit for all-cause and cardiovascular death observed in the sevelamer group of our cohort, after adjusting for calcium-based binders use (instead comparing to calcium-based binders use) constitutes the most original finding of this study. Moreover, the comparative effect between sevelamer against calcium-based binders on survival have been already addressed in the only randomized clinical trial aimed to evaluate all-cause mortality as the primary end point in NDD-CKD patients. As we note in the Discussion section, in that study Di Iorio et al. [36] observed a lower mortality rate among patients receiving sevelamer Vs. calcium carbonate.
Lastly, as we also noted in the Discussion section, use of calcium-based phosphate binders did not predict death in our cohort of patients (data not shown). Altogether, we think that this lack of survival benefit observed among patients receiving calcium-based binders might have a very relative interest, while minimizing the originality of our study results.
Comment 2. Numerous experimental, observational studies, and clinical trials have highlighted Sevelamer's multifaceted effects beyond controlling hyperphosphatemia. These effects encompass actions on inflammation, oxidative stress, lipid profile, atherogenesis, vascular calcification, endothelial dysfunction, and reduction of uremic toxins. Acknowledging these pleiotropic effects in the discussion, along with its ability to bind bile salts—a factor potentially contributing to benefits such as reduced low-density lipoprotein cholesterol and plasma glucose levels (mentioned in lines 247 to 248)—could enrich the overall understanding of Sevelamer's comprehensive impact on cardiovascular morbidity and mortality in chronic kidney disease patients.
The authors agree with this comment. In this new version of the manuscript, we have pointed out the potential pleiotropic effects of sevelamer beyond controlling hyperphosphatemia. Accordingly, we have added the following text to the Discussion section, along with four new citations:
“Additionally, sevelamer may produce further benefits through many pleiotropic actions on lipid profile, inflammatory markers, uremic toxins and blood glucose levels in CKD patients [40, 41]. Based on its ability to bind bile salts, sevelamer has been demonstrated to reduce low-density lipoprotein cholesterol and plasma glucose levels [42-44]. These effects have been recently summarized in a meta-analysis including 44 studies for qualitative analysis and 28 reports for quantitative analysis, in which sevelamer showed a significant reduction in cholesterol levels as compared to calcium-based phosphate binders, with a decrease in glycated hemoglobin levels in sevelamer-treated patients [40]. Anti-inflammatory effects for sevelamer-based products have also been well demonstrated. In a randomized, controlled, open-label, crossover trial involving 53 patients with NDD-CKD, Ruggiero et al. showed that sevelamer reduced CRP, glycated hemoglobin, and total and low-density lipoprotein cholesterol levels and increased high-density lipoprotein cholesterol levels without affecting GFR, proteinuria, blood pressure, or levels of fibroblast growth factor 23, klotho, intact parathyroid hormone or serum vitamin D [45]. Decreases in biomarkers of oxidative stress as well as in uremic toxins as p-cresyl sulfate are additional benefits observed after sevelamer treatment [41,46].”
- Basutkar RS, Varghese R, Mathew NK, Sankar Indira P, Viswanathan B, Sivasankaran P. Systematic review and meta-analysis of potential pleiotropic effects of sevelamer in chronic kidney disease: Beyond phosphate control. Nephrology (Carlton). 2022 Apr;27(4):337-354. doi: 10.1111/nep.14011. Epub 2021 Dec 27. PMID: 34882904.
- Lin YF, Chien CT, Kan WC, Chen YM, Chu TS, Hung KY, Tsai TJ, Wu KD, Wu MS. Pleiotropic effects of sevelamer beyond phosphate binding in end-stage renal disease patients: a randomized, open-label, parallel-group study. Clin Drug Investig. 2011;31(4):257-67. doi: 10.2165/11539120-000000000-00000. PMID: 21299254.
- Ruggiero B, Trillini M, Tartaglione L, Rotondi S, Perticucci E, Tripepi R, Aparicio C, Lecchi V, Perna A, Peraro F, Villa D, Ferrari S, Cannata A, Mazzaferro S, Mallamaci F, Zoccali C, Bellasi A, Cozzolino M, Remuzzi G, Ruggenenti P, Kohan DE; ANSWER Study Organization. Effects of Sevelamer Carbonate in Patients With CKD and Proteinuria: The ANSWER Randomized Trial. Am J Kidney Dis. 2019 Sep;74(3):338-350. doi: 10.1053/j.ajkd.2019.01.029. Epub 2019 Apr 23. PMID: 31027883.
- Lin CJ, Pan CF, Chuang CK, Liu HL, Huang SF, Chen HH, Wu CJ. Effects of Sevelamer Hydrochloride on Uremic Toxins Serum Indoxyl Sulfate and P-Cresyl Sulfate in Hemodialysis Patients. J Clin Med Res. 2017 Sep;9(9):765-770. doi: 10.14740/jocmr1803e. Epub 2017 Jul 27. PMID: 28811853; PMCID: PMC5544481.
Round 2
Reviewer 1 Report
Comments and Suggestions for Authors
I have no further comments